# One-Step Fabrication of Highly Sensitive Tris(2,2′-bipyridyl)ruthenium(II) Electrogenerated Chemiluminescence Sensor Based on Graphene-Titania-Nafion Composite Film

**DOI:** 10.3390/s22083064

**Published:** 2022-04-15

**Authors:** Sang Jung Lee, Don Hui Lee, Won-Yong Lee

**Affiliations:** Department of Chemistry, Yonsei University, Seoul 03722, Korea; sangjung@yonsei.ac.kr (S.J.L.); ttlfqptmxpfm@naver.com (D.H.L.)

**Keywords:** electrogenerated chemiluminescence, tris(2,2′-bipyridyl)ruthenium(II), graphene, NADH, ECL sensor

## Abstract

A highly sensitive tris(2,2′-bipyridyl)ruthenium(II) (Ru(bpy)_3_^2+^) electrogenerated chemiluminescence (ECL) sensor based on a graphene-titania-Nafion composite film has been prepared in a simple one-step manner. In the present work, a highly concentrated 0.1 M Ru(bpy)_3_^2+^ solution was mixed with an as-prepared graphene-titania-Nafion composite solution (1:20, *v*/*v*), and then a small aliquot (2 µL) of the resulting mixture solution was cast on a glassy carbon electrode surface. This one-step process for the construction of an ECL sensor shortens the fabrication time and leads to reproducible ECL signals. Due to the synergistic effect of conductive graphene and mesoporous sol-gel derived titania-Nafion composite, the present ECL sensor leads to a highly sensitive detection of tripropylamine from 1.0 × 10^−8^ M to 2.0 × 10^−3^ M with a detection limit of 0.8 nM (S/N = 3), which is lower in comparison to that of the ECL sensor based on the corresponding ECL sensor based on the titania-Nafion composite containing carbon nanotube. The present ECL sensor also shows a good response for nicotinamide adenine dinucleotide hydrogen (NADH) from 1.0 × 10^−6^ M to 1.0 × 10^−3^ M with a detection limit of 0.4 µM (S/N = 3). Thus, the present ECL sensor can offer potential benefits in the development of dehydrogenase-based biosensors.

## 1. Introduction

Electrogenerated chemiluminescence (ECL) is the light emission generated from a high energy electron transfer reaction between electrogenerated species on electrode surfaces [1]. Since ECL has definite advantages including low background signal and high sensitivity, it is widely used as a powerful transduction method in chemical analysis, immunoassay, and DNA probe assay [2,3]. Among several ECL luminophores, tris(2,2′-bipyridyl)ruthenium(II) (Ru(bpy)_3_^2+^) has been most widely used to solve real analytical problems because of its inherent chemical stability and high quantum efficiency. Moreover, since Ru(bpy)_3_^2+^ can be regenerated in its ECL emission process, the immobilization of Ru(bpy)_3_^2+^ on electrode surfaces facilitates the real application of ECL-based sensors and biosensors in chemical and biochemical analysis. Therefore, a large number of methods have been developed to immobilize Ru(bpy)_3_^2+^ or its derivatives on electrode surfaces, including the physical incorporation of Ru(bpy)_3_^2+^ into various ionomer-based composite films [4,5,6,7,8,9], chemically anchored Ru(bpy)_3_^2+^ derivatives in organic or inorganic polymeric systems [10,11], a self-assembled Ru(bpy)_3_^2+^ derivative [12], and inorganic particle- Ru(bpy)_3_^2+^ multilayer films [13,14,15]. Despite their good ECL performances, there have been more demands to increase the response and long-term stability of the Ru(bpy)_3_^2+^ ECL sensors. Over the last decade, several nanomaterials such as carbon nanotube (CNT) and graphene have been introduced in the Ru(bpy)_3_^2+^ ECL sensors in order to improve their analytical performance [16,17,18,19,20,21,22,23,24,25]. In particular, graphene has attracted immense attention in ECL detection systems because it can offer distinct benefits including excellent conductivity, high mechanical strength, and good biocompatibility [26]. Up to now, several graphene-based ECL sensors have been developed to detect tripropylamine (TPA) [19,20,22,23], oxalate [19,20], DNA [21], ethanol [24], and α-fetoprotein [25]. All the reported graphene-based Ru(bpy)_3_^2+^ ECL sensors so far were prepared in many different ways for the immobilization of Ru(bpy)_3_^2+^-graphene on various working electrode surfaces. For example, Xu’s group reported on a conventional two-step method for the preparation of a graphene-based Ru(bpy)_3_^2+^ ECL sensor, in which a small aliquot of graphene-Nafion composite solution was drop-casted onto a GC electrode surface, and then the graphene-Nafion composite-modified GC electrode was dipped in Ru(bpy)_3_^2+^ solution to immobilize Ru(bpy)_3_^2+^ by an ion-exchange process [19]. The resulting Ru(bpy)_3_^2+^ ECL sensor exhibited a good response towards TPA with a detection limit of 50 nM. Gao and coworkers reported on a three-step method for the construction of a graphene-based Ru(bpy)_3_^2+^ ECL sensor without the incorporation of auxiliary mediums [20]. They first drop-cast a small aliquot of Ru(bpy)_3_^2+^ solution on a GC electrode surface, and then a small aliquot of graphene oxide solution was dropped over the top of the Ru(bpy)_3_^2+^ film. Finally, the graphene oxide film was reduced to graphene via in situ wet-chemical reduction under the reductant solution. Despite the relatively complicated and long fabrication process over 16 h, the ECL sensor showed a good response towards TPA with a detection limit of 5.0 nM. Wang’s group reported on the paper-based Ru(bpy)_3_^2+^ ECL sensors, in which the composite film of poly(sodium 4-styrenesulfonate) functionalized graphene (PSSG)/Nafion was screen printed first, and then ion-exchange immobilization of Ru(bpy)_3_^2+^ on the composite-modified electrode was carried out [21]. In addition, a PSSG/carbon paste composite containing highly concentrated Ru(bpy)_3_^2+^ was screen printed in one-step manner [22]. These ECL sensors also showed a good response towards TPA with a detection limit of 5.0 nM. Qian and coworkers reported on the porous graphene-based Ru(bpy)_3_^2+^ ECL sensor, in which a three-dimensional graphene with interpenetrating porous structures (3D-pGR) was first produced from freeze-drying dispersions containing graphene oxide followed by calcination, and then a small aliquot of 3D-pGR-Nafion composite solution was drop-casted on a GC electrode, and the resulting composite-modified GC electrode was immersed in Ru(bpy)_3_^2+^ solution [23]. The resulting ECL sensor showed a good response towards TPA with a detection limit of 0.4 nM. Although all the reported graphene-based ECL sensors exhibited good analytical performance, they still suffered from complex and troublesome modification steps of the graphene surface or time-consuming multi-step immobilization processes.

Therefore, we report herein on a simple and efficient one-step method for the fabrication of a Ru(bpy)_3_^2+^ ECL sensor based on a mesoporous titania-Nafion composite film containing graphene. Previously, it was found that the use of a mesoporous and biocompatible composite film of sol-gel derived titania (TiO_2_) and Nafion in the Ru(bpy)_3_^2+^ ECL sensor greatly improved the sensitivity and long-term stability of the resulting ECL sensor [8]. Therefore, the present Ru(bpy)_3_^2+^ ECL sensor based on the titania-Nafion composite film containing graphene can combine the synergistic effect of the conductive graphene and highly porous titania-Nafion composite film, leading to a highly sensitive detection of TPA and other coreactants as target analytes. In the present work, the analytical performance of the present Ru(bpy)_3_^2+^ ECL sensor based on the graphene-titania-Nafion composite film was compared to that obtained with the Ru(bpy)_3_^2+^ ECL sensor based on the carbon nanotube (CNT) within the identical titania-Nafion composite film. To the best of our knowledge, this is the first report on the comparison of the analytical performance of the Ru(bpy)_3_^2+^ ECL sensors based on graphene vs. CNT. The present Ru(bpy)_3_^2+^ ECL sensor was also applied to the sensitive determination of NADH in human serum samples. Compared with the previous reports on the graphene-based ECL sensors [19,20,21,22,23], the present one-step process for the fabrication of the ECL sensor based on a graphene-titania-Nafion composite film is facile, rapid, and cost-effective with good improvements in sensitivity, long-term stability, and reproducibility.

## 2. Materials and Methods

### 2.1. Reagents

Titanium(IV) isopropoxide (Ti(OR)_4_, R = CH(CH_3_)_2_, 99.999%), Nafion (perfluorinated ion exchange resin, 5 wt.% in lower aliphatic alcohols and water contains 15~20% water), tripropylamine (TPA, 99%), tris(2,2′-bibpyridyl)dichlororuthenium(II) hexahydrate (Ru(bpy)_3_Cl_2_·6H2O, 98%), and beta-nicotinamide adenine dinucleotide reduced disodium salt hydrate (β-NADH) and human serum were purchased from Sigma-Aldrich. 2-Propanol (≥99.9%) (Fluka, Germany) and hydrochloric acid (35–37%) (Samchun, Korea) were used to prepare sol-gel derived titania. Graphene nanoplatelets were obtained from Cheap Tubes Inc. (Grafton, VT, USA). Carbon nanotube, multi-walled (>99% MWCNT, diameter: 6~13 nm, and length: 2.5~20 µm) was obtained from Sigma-Aldrich. All chemicals were used as received without any further purification. All experimental solutions were prepared using 0.05 M phosphate buffer solution at pH 7. Water for all solutions was purified using a Milli-Q water purification system (Bedford, MA, USA).

### 2.2. Instrumentation

Electrochemical and ECL experiments were carried out with a conventional three-electrode configuration in a 15 mL quartz electrochemical cell. Bare glassy carbon (GC) electrode (geometrical area = 0.071 cm^2^) modified with Ru(bpy)_3_^2+^-graphene-titania-Nafion composite film was used as the working electrode, while a platinum wire and Ag/AgCl (3 M NaCl) were used as the counter and reference electrode, respectively. A potentiostat of EG&G 263A (Princeton, NJ, USA) was used in order to perform the cyclic voltammetry (CV) and ECL experiments. The surface coverage, Г, was determined by integrating the background-corrected cyclic voltammogram obtained at the Ru(bpy)_3_^2+^ immobilized at the composite-modified GC electrodes. In the electrochemical impedance spectroscopy, the impedance spectra were obtained in the frequency range of 10 mHz to 100 kHz at an applied potential of 0.2 V using an alternating voltage of 5.0 mV in 5.0 mM K_3_Fe(CN)_6_/K_4_Fe(CN)_6_ (1:1, *v*/*v*) solution. ZSimpWin program (Princeton Applied Research, Oak Ridge, TN, USA) was used to fit the EIS experimental results into a proper equivalent circuit. In ECL experiments, a conventional quartz cell equipped with three electrodes was positioned in front of the PMT window in order to collect the ECL emission. The light emissions were counted with Hamamatsu Photonics H9319-12 photon counting module (Shizuoka, Japan) as previously described [8]. All ECL experiments were carried out in a lightproof black box at room temperature.

### 2.3. Preparation of ECL Sensor

The titania (TiO_2_) sol was prepared by hydrolysis and condensation by mixing 1 mL of 0.15 M Ti(OC_4_H_9_)_4_ dissolved in 2-propanol and 5 mL of deionized water as described in a previous report [8]. Around 7 µL of 0.1 M HCl solution was added to the above solution as a catalyst. The solution was vigorously stirred for 1 h at room temperature until it became transparent. The prepared titania sol was mixed with Nafion (5 wt.%) to form a homogeneous titania-Nafion composite solution (1:1, *v*/*v*). Graphene was dispersed in the as-prepared titania-Nafion composite solution (1.25 mg/mL) in an ultrasonic bath for around 1 h as previously reported [27]. In order to prepare the present ECL sensor, 0.1 M Ru(bpy)_3_^2+^ aqueous solution was added in the as-prepared graphene-titania-Nafion composite solution (1:20, *v*/*v*), and then 2 µL aliquot of the resulting composite solution was drop-casted on a GC electrode surface. The ECL sensor was dried in the air for 30 min in a lightproof environment, and then placed in the 0.05 M phosphate buffer solution at pH 7.0 for 15 min before use. Prior to the casting of the composite solution, a bare GC electrode was polished using alumina powder (0.05 µm) on a polishing cloth and then sonicated in ethanol. Then, the GC electrode was washed with DI water, and dry-aged in the air under the room temperature.

### 2.4. ECL Experiments

For ECL measurements, the as-prepared ECL sensor was immersed in a solution of analyte, and a cyclic potential was applied to the ECL sensor from an initial potential of 0.0 V to a high of +1.35 V and back to 0.0 V with a scan rate of 100 mV/s vs. Ag/AgCl (3 M NaCl). Human serum was chosen as a real sample. Prior to ECL experiments, human serum sample was 100-fold diluted with 50 mM phosphate buffer solution at pH 7.0 without any filtration step. The recovered concentrations of the spiked NADH solutions were derived from the standard addition method.

## 3. Results and Discussion

### 3.1. Characterization of the One-Step Prepared ECL Sensor

In the present work, graphene was incorporated in the composite of sol-gel derived titania and Nafion. As previously reported [27], graphene was well dispersed in the composite solution composed of titania and Nafion (1:1, *v*/*v*). The present Ru(bpy)_3_^2+^ ECL sensor was prepared in one-step manner, in which a highly concentrated 0.1 M Ru(bpy)_3_^2+^ solution was mixed with the as-prepared graphene-titania-Nafion composite solution (1:20, *v*/*v*), and then a 2 µL aliquot of the resulting composite solution was drop-casted on a pre-cleaned GC electrode surface. The Ru(bpy)_3_^2+^-incorporated graphene-titania-Nafion composite was well adhered on the GC electrode and the coated composite film did not come off during the electrochemical and ECL experiments. In order to characterize the morphological features, a scanning electron micrograph (SEM) image of the one-step prepared ECL sensor based on the graphene-titania-Nafion composite film was obtained and compared to that of the corresponding ECL sensor based on the titania-Nafion composite film. As shown in Figure 1, the titania-Nafion composite film is highly porous and microscopically rough, similar to the previous report [27], while the incorporation of graphene within the titania-Nafion composite forms a finer and more uniform film from the lamellar structures of graphenes.

### 3.2. Electrochemical Behavior

Electrochemical behaviors of the present ECL sensor were studied using cyclic voltammetry (CV) and electrochemical impedance spectroscopy (EIS). Figure 2 shows typical cyclic voltammograms of Ru(bpy)_3_^2+^ immobilized in three different composite-modified GC electrodes (geometrical area = 0.071 cm^2^) in a 0.05 M phosphate buffer solution (PBS) at pH 7.0. The voltammogram obtained by the present Ru(bpy)_3_^2+^ ECL sensor based on the graphene-titania-Nafion composite (red line) was similar in shape to that obtained by the ECL sensor based on the titania-Nafion composite without graphene (black line). The oxidation current obtained by the Ru(bpy)_3_^2+^ ECL sensor based on the graphene-titania-Nafion composite (2.29 × 10^−5^ A) was about 15% greater than that obtained by the Ru(bpy)_3_^2+^ ECL sensor based on the titania-Nafion composite (blue line) without graphene (2.00 × 10^−5^ A), and was also 9.5% higher than that obtained by the Ru(bpy)_3_^2+^ ECL sensor based on the CNT-titania-Nafion composite (2.09 × 10^−5^ A).

In addition, the surface coverage, Г, was determined for the Ru(bpy)_3_^2+^ ECL sensors by integrating the background-corrected cyclic voltammogram of Ru(bpy)_3_^2+^ obtained at a scan rate of 1.0 mv/s with the following equation
Г = Q/nFA
where Q, n, F, and A represent the charge involved in the reaction, the number of electrons transferred in the redox reaction, Faraday*’*s constant, and electrode area, respectively. The calculated Г of the ECL sensor based on the graphene-titania-Nafion composite was 4.9 × 10*^−^*^8^ mol/cm^2^, which was 75% higher than that of the ECL sensor based on the titania-Nafion composite without graphene (2.8 × 10*^−^*^8^ mol/cm^2^). It was also 36% higher than that of the ECL sensor based on the CNT-titania-Nafion composite (3.6 × 10*^−^*^8^ mol/cm^2^). The increased amount of Ru(bpy)_3_^2+^ in the present ECL sensor based on the graphene-titania-Nafion composite could be attributed to the large surface area, hydrophobicity, and high π conjugation of graphene within the composite film, which leads to the effective immobilization of Ru(bpy)_3_^2+^ via hydrophobic π–π interactions between graphene and bipyridyl ligands in the Ru(bpy)_3_^2+^ complex. The anodic peak current of Ru(bpy)_3_^2+^ was proportional to the square root of the scan rate (*v*^1/2^) over the range of 10~200 mV/s, indicating that the immobilized Ru(bpy)_3_^2+^ within the graphene-titania-Nafion composite undergoes a diffusion process within the composite films on the GC electrode as reported previously [8,17,19]. In order to estimate the mass transport rate within the graphene-titania-Nafion composite-modified GC electrode, the apparent diffusion coefficient, *D_app_*, for Ru(bpy)_3_^2+^ for the present ECL sensor was determined by the Randles–Sevick equation according to the previous report [8]. The calculated *D_app_* for Ru(bpy)_3_^2+^ from the present ECL sensor was around 2.93 × 10*^−^*^10^ cm^2^/s, which was slightly smaller than that measured by the ECL sensor based on the titania-Nafion composite (4.1 × 10*^−^*^9^ cm^2^/s) [8], but was quite analogous to that for Ru(bpy)_3_^2+^ from the CNT-titania-Nafion composite-modified GC electrodes (3.01 × 10*^−^*^10^ cm^2^/s). The present experimental result indicates that the incorporation of graphene within the titania-Nafion composite film slightly decreases the mass transport rate within the composite film on the GC electrode surface, which is similar to the previously reported result for Ru(bpy)_3_^2+^ with the CNT-titania-Nafion composite film [17].

The electrochemical impedance spectroscopy (EIS) measurements were carried out to obtain the information on the interfacial property at the graphene-titania-Nafion composite-modified GC electrode surface. As a redox probe, [Fe(CN)_6_]^3-/4-^ was used in the present study as similar to other previous EIS studies for graphene-incorporated composite-modified electrodes [21,25]. As shown in Figure 3, the EIS experimental results were fitted into the R(QR)W equivalent circuit model (inset of Figure 3) using ZSimpWin program, where R_s_, Q_dl_, and W represent the solution resistance, double-layer capacitance, and Warburg impedance, respectively [27]. The semicircle diameter in Nyquist plots corresponds to the electron transfer resistance, R_et_, which is related to the electron transfer kinetics of the redox probe [Fe(CN)_6_]^3-/4-^ at the composite-modified GC electrode interface.

The R_et_ obtained by the graphene-titania-Nafion composite-modified GC electrode was 2.78 × 10^3^ Ω, which was much smaller than that obtained by the titania-Nafion composite-modified GC electrode (1.64 × 10^6^ Ω). This result clearly demonstrates that the introduction of electrically conductive graphene into the titania-Nafion composite can improve the electron transfer of the [Fe(CN)_6_]^3-/4-^ redox probe at the composite-modified GC electrode interface. In addition, the R_et_ obtained by the graphene-titania-Nafion composite-modified GC electrode was also smaller than that obtained by the CNT-titania-Nafion composite-modified GC electrode (2.45 × 10^5^ Ω), confirming that graphene is a more effective carbon nanomaterial than CNT for the construction of the solid-state ECL sensors.

### 3.3. ECL Behavior

The ECL behavior of the present Ru(bpy)_3_^2+^ ECL sensor based on the graphene-titania-Nafion composite film was first examined with TPA as a coreactant. As shown in Figure 4, cyclic voltammograms and subsequent ECL-potential plots were simultaneously acquired by the present ECL sensor (geometrical area = 0.071 cm^2^) with (solid line) and without (dashed line) the presence of 0.5 mM TPA in 0.05 M PBS (pH 7.0).

As 0.5 mM TPA was introduced in 0.05 M PBS, the oxidation current of Ru(bpy)_3_^2+^ was increased, while its reduction current was rather slightly decreased (Figure 4A). Furthermore, the ECL signal was significantly increased and the onset potential of ECL emission was almost coherent with the oxidation potential of Ru(bpy)_3_^2+^ immobilized in the graphene-titania-Nafion composite film at around + 1.0~1.1 V (vs. Ag/AgCl, 3 M NaCl) (Figure 4B). The ECL intensity obtained by the present ECL sensor was 2-fold greater than that obtained by the ECL sensor based on the corresponding composite without graphene, and also was 25% higher than that from the ECL sensor based on the corresponding composite with CNT. The increased ECL response by the present ECL sensor could be attributed to three reasons. First, the increased amount of Ru(bpy)_3_^2+^ immobilized at the graphene-titania-Nafion composite, as discussed in the earlier CV results, might lead to an increase in the ECL response for TPA because the ECL response is proportional to not only the concentration of coreactant but also the amount of Ru(bpy)_3_^2+^ immobilized on the ECL sensor [19]. Second, the introduction of hydrophobic graphene in the titania-Nafion composite films could result in the increased oxidation rate of hydrophobic TPA and thus leads to the increased ECL response. This result corresponds to the previous discovery that the hydrophobicity of an electrode surface strongly affects the ECL response for TPA [17,28]. Finally, the enhancement in the rates of electron transfer and diffusion in the graphene-titania-Nafion composite film could also lead to the increased ECL response.

The reproducibility of the present one-step prepared Ru(bpy)_3_^2+^ ECL sensor has been tested for the repetitive measurements of 0.5 mM TPA solution at a scan rate of 100 mV/s. As shown in Figure 5, the relative standard deviation (RSD) in 30 consecutive measurements was 3.6%, indicating good reproducibility of the present ECL sensor. In addition, the effect of the amount of graphene incorporated into the titania-Nafion composite on the ECL response for 0.5 mM TPA was studied at a scan rate of 100 mV/s. As shown in Figure 6, the ECL intensity increased sharply as the graphene amount increased from 0.25 mg/mL to 1.25 mg/mL. The result clearly indicates that the electron transfer within the composite and the electrochemical oxidation of TPA on the ECL sensor increases as more graphene is incorporated within the composite. Meanwhile, a further increase in the graphene amount in the composite leads to a slow decrease in the ECL response and also a much less reproducible ECL response, which can be ascribed to the poor dispersion and strong aggregation of graphene in the composite film. Therefore, the ECL sensor was prepared with the optimal amount of graphene of 1.25 mg/mL within the titania-Nafion composite solution in all subsequent experiments in order to ensure a strong and reproducible ECL response.

The ECL response (*I_ECL_*) towards the concentration of TPA in the logarithmic scale was observed with the present ECL sensor in two ranges. As illustrated in Figure 6, the first linear range was from 1.0 × 10*^−^*^8^ M to 5.0 × 10*^−^*^7^ M with the regression equation (*I_ECL_* = 7.01 × 10^10^ [TPA] (M) + 50,100; *R^2^* = 0.994), and the second range from 5.0 × 10*^−^*^7^ M to 2.0 × 10*^−^*^3^ M with the regression equation (log *I_ECL_* = (0.591) log[TPA] (M) + 8.649; *R^2^* = 0.999). The limit of detection (LOD) for TPA was found to be 8.2 × 10*^−^*^10^ M (S/N = 3). Furthermore, the present ECL sensor showed good reproducibility for the detection of TPA. The RSD in the repeated ECL measurements were between 0.49% and 9.98% (n = 3) for TPA with concentrations from 1.0 × 10*^−^*^8^ M to 2.0 × 10*^−^*^3^ M as shown in Figure 7. As summarized in Table 1, the linear dynamic range and LOD for TPA obtained by the present ECL sensor is comparable or slightly superior to those previously reported by graphene-based ECL sensors [16,17,19,20,21,22,23]. In particular, the LOD for TPA for the present ECL sensor based on the graphene-titania-Nafion composite was two orders of magnitude lower than that obtained for the two-step prepared ECL sensor based on the titania-Nafion composite without graphene [8], and was also one order of magnitude lower than that for the two-step prepared ECL sensor based on the CNT-titania-Nafion composite [17].

### 3.4. Selectivity and Stability

The response selectivity of the present ECL sensor towards various analytes was studied. As listed in Table 2, the ECL response of the present ECL sensor was strongly dependent upon the charge and hydrophobicity of analytes. For example, the ECL responses of the present ECL sensor were extremely small (less than 0.25%) for anionic analytes (oxalate and ascorbic acid) relative to that of TPA, which is ascribed to the charge repulsion between the negatively charged composite and anionic analytes, leading to unfavorable diffusion of the analytes towards the ECL sensor. Meanwhile, the present ECL sensor exhibited much stronger responses for hydrophobic analytes such as NADH (8.7%), promazine (10.4%), and erythromycin (17.8%) than those (NADH: 1.9%, promazine: 3.5%, erythromycin: 6.6%) obtained by the conventional solution phase ECL measurement at a bare GC electrode in the presence of 1.0 mM Ru(bpy_3_)^2+^ solution as previously reported [8]. Furthermore, the present ECL sensor did not respond to dopamine. Therefore, the present ECL sensor based on the graphene-titania-Nafion composite film could be preferably used for monitoring the concentration of antibiotics such as erythromycin in biological fluids (urine and blood) and for detecting NADH or substrates for dehydrogenase-based biosensors that enzymatically liberate NADH [29,30,31,32,33].

The long-term storage stability of the Ru(bpy_3_)^2+^ ECL sensor based on the graphene-titania-Nafion composite was studied over a 3-week period by monitoring its ECL response for 0.5 mM TPA with intermittent measurements after storage in the 0.05 M phosphate buffer solution at room temperature. The ECL response for TPA decreased to 83% of its initial value in 3 weeks. Before and after the long-term stability test, a noticeable change in the morphology of the composite film was not observed and the coated composite film on the GC electrode also did not come off. The stability of the present ECL sensor is far superior to the two-step prepared ECL sensors based on Nafion, or ionomer-sol-gel metal oxide films such as silica and titania without the incorporation of CNT or graphene [5,6,7,8]. The good stability of the present ECL sensor could be attributed to the strong hydrophobic π–π interactions between Ru(bpy_3_)^2+^ and graphene, preventing Ru(bpy_3_)^2+^ from leaking out of the graphene-titania-Nafion composite film [16]. Furthermore, the incorporation of graphene might delay the migration of the Ru(bpy_3_)^2+^ into the electrochemically inactive hydrophobic region of Nafion, as observed in the previous reports with the ECL sensors based on graphene-Nafion [19] and CNT-titania-Nafion [17].

### 3.5. ECL Detection of NADH

In order to test the feasibility of the present Ru(bpy)_3_^2+^ ECL sensor based on the graphene-titania-Nafion composite in a real biological sample, NADH was selected as a representative analyte. Since NADH plays as a coreactant in the oxidation-reduction pathway of the Ru(bpy)_3_^2+^ ECL system [4], its ECL behavior is almost identical to that of TPA. The relative ECL response of NADH obtained by the present ECL sensor was determined to be 8.7% compared to that of TPA under the identical experimental conditions. The quantitative ECL detection of NADH was performed using the present ECL sensor by cycling the working electrode potential between 0.0 and +1.35 V (vs. Ag/AgCl, 3 M NaCl) at a scan rate of 100 mV/s in 0.05 M PBS at pH 7.0. The ECL response (*I_ECL_*) towards the concentration of NADH in the logarithmic scale was observed in two ranges. As illustrated in Figure 8, the first linear range was 1.0 × 10^−6^ M to 1.0 × 10^−5^ M with the regression equation (*I_ECL_* = 2.37× 10^8^ [NADH] (M) + 2332; *R^2^* = 0.982), and the second range from 1.0 × 10^−5^ M to 1.0 × 10^−3^ M with the regression equation (*I_ECL_* = 10,076 log[NADH] (M) + 54,613; *R^2^* = 0.970). The LOD for NADH was calculated to be 3.51 × 10^−7^ M (S/N = 3). The linear range and LOD for NADH obtained with the present ECL sensor were analogous to those obtained with previous electrochemical and ECL detection methods [31,32,33]. Furthermore, the ECL responses observed for the present ECL sensor were quite reproducible with RSD in the range from 1.04% to 4.61% (n = 3) with NADH concentrations from 1.0 × 10^−6^ M to 1.0 × 10^−3^ M. This experimental result clearly implies that the present ECL sensor based on the graphene-titania-Nafion has great potential as a transduction platform in the development of dehydrogenase-based enzyme biosensors because the present ECL sensor does

Not cause the contamination of the electrode surface due to undesirable interference reactions commonly encountered in the electrochemical oxidation of NADH [32]. In addition, the ECL biosensor based on the present graphene-titania-Nafion composite could possibly exhibit good long-term stability because of the excellent biocompatibility of the present titania-Nafion composite.

The real applicability of the present Ru(bpy)_3_^2+^ ECL sensor for NADH was tested via a recovery test in the human serum sample. A conventional standard addition method was carried out at least three times to derive the recovered concentrations of NADH in human serum samples. The test results, as listed in Table 3, were good with 102.4% and 104.2% (n = 3) for 100 µM and 400 µM NADH spiked in human serum samples, respectively. These results strongly suggest a promising applicability of the present ECL sensor as a highly sensitive transduction system in the development of dehydrogenase-based biosensors and bioassays in biological fluids.

## 4. Conclusions

In the present work, a highly sensitive ECL sensor based on Ru(bpy)_3_^2+^ immobilized on the graphene-titania-Nafion composite film has been constructed in a simple one-step manner. This one-step approach for the construction of an ECL sensor shortens the fabrication time and leads to more reproducible ECL signals in comparison to the conventional two-step process. By virtue of the synergistic effect of the conductive graphene and mesoporous titania-Nafion composite, the present Ru(bpy)_3_^2+^ ECL sensor based on the graphene-titania-Nafion composite film allows a highly sensitive detection of TPA from 1.0 × 10**^−^**^8^ M to 2.0 × 10**^−^**^3^ M with a detection limit of 0.8 nM (S/N = 3), which is one order of magnitude lower in comparison to previously reported ECL sensors based on CNT. Moreover, the present ECL sensor has been successfully applied to the determination of NADH in human serum samples with good recovery. Therefore, the present Ru(bpy)_3_^2+^ ECL sensor offers a good transduction platform in the quantitative analysis of coreactants, immunoassay, DNA probe assay, and dehydrogenase-based biosensors for the detection of species such as glucose, ethanol, and lactate.

## Figures and Tables

**Figure 1 sensors-22-03064-f001:**
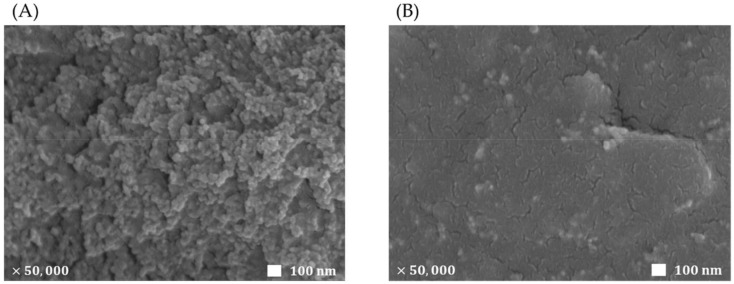
SEM images of one-step prepared ECL sensors based on titania-Nafion (**A**) and graphene-titania-Nafion (**B**) composite films (graphene: 1.25 mg/mL). White bars in figures are equivalent to 100 nm.

**Figure 2 sensors-22-03064-f002:**
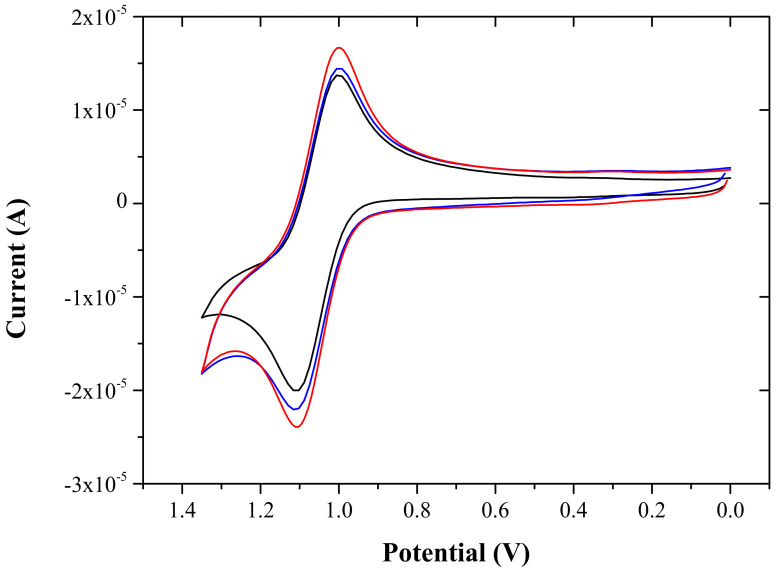
Cyclic voltammograms of 0.5 mM Ru(bpy)_3_^2+^ in 0.05 M PBS (pH 7.0) from the titania-Nafion composite-modified (black), graphene-titania-Nafion composite-modified (red), and CNT-titania-Nafion composite-modified (blue) GC electrodes with a scan rate of 100 mV/s.

**Figure 3 sensors-22-03064-f003:**
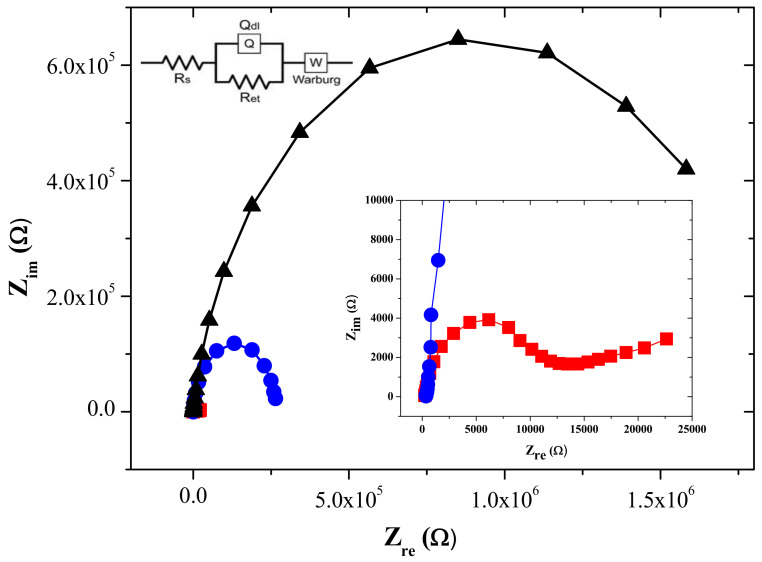
Nyquist plots for the impedance measurements in the presence of 5.0 mM K_3_Fe(CN)_6_/K_4_Fe(CN)_6_ in 0.05 M PBS (pH 7.0) by the titania-Nafion composite-modified (black), graphene-titania-Nafion composite-modified (red), and CNT-titania-Nafion composite-modified (blue) GC electrode. Inset: four-component equivalent circuit R(QR)W, R_et_: electron transfer resistance, R_s_: solution resistance, Q_dl_: double-layer capacitance, W: Warburg impedance.

**Figure 4 sensors-22-03064-f004:**
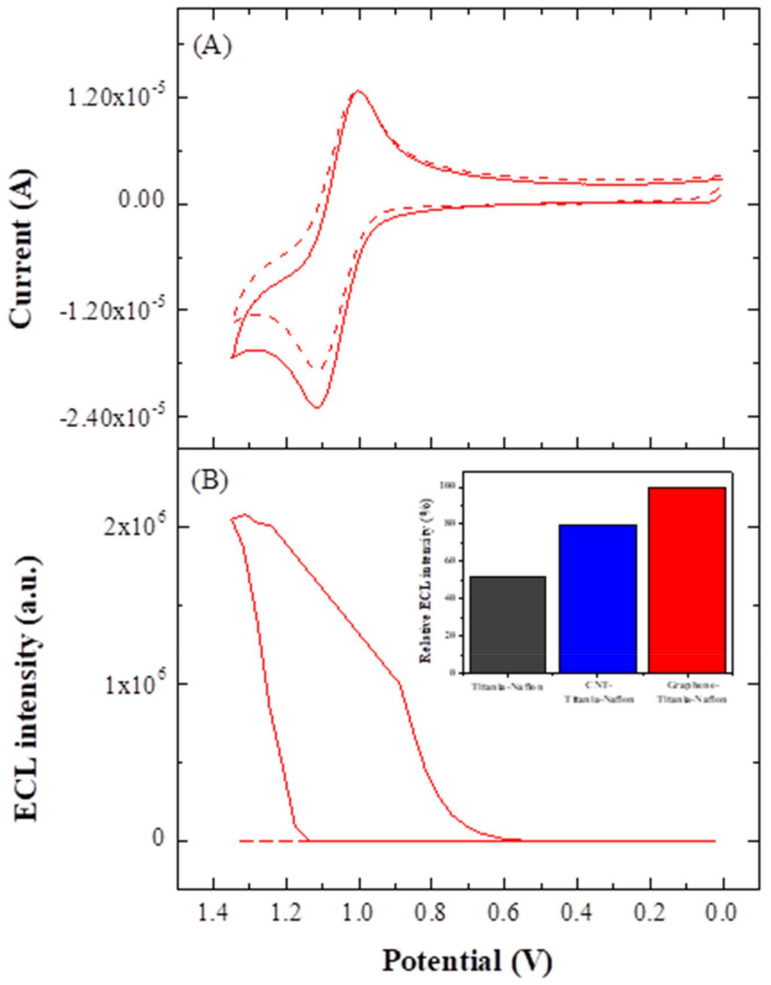
Cyclic voltammograms (**A**) and ECL-potential plots (**B**) acquired by the present ECL sensor based on graphene-titania-Nafion composite without (dashed line) and with (solid line) the presence of 0.5 mM TPA in 0.05 M PBS (scan rate = 100 mV/s). Inset in Figure 4B: Relative ECL response obtained by the ECL sensors based on titania-Nafion composite (black), graphene-titania-Nafion (red), and CNT-titania-Nafion composite (blue) under the same experimental conditions.

**Figure 5 sensors-22-03064-f005:**
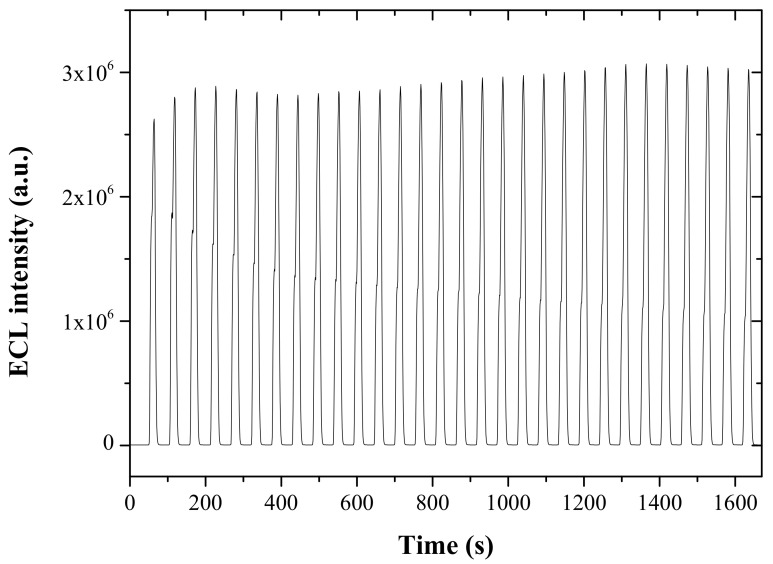
ECL-time responses with 30 consecutive measurements for 0.5 mM TPA in 0.05 PBS (pH 7.0) obtained by the one-step prepared ECL sensor based on the graphene-titania-Nafion composite film with a scan rate of 100 mV/s.

**Figure 6 sensors-22-03064-f006:**
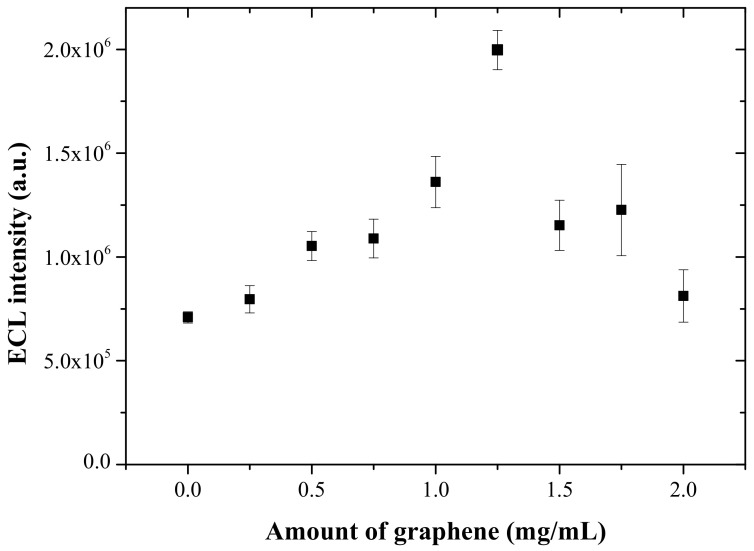
Effect of the graphene amount in the titania-Nafion composite film on ECL intensity. The ECL intensity was obtained for 0.5 mM TPA in 0.05 M PBS (pH 7.0) with a scan rate of 100 mV/s.

**Figure 7 sensors-22-03064-f007:**
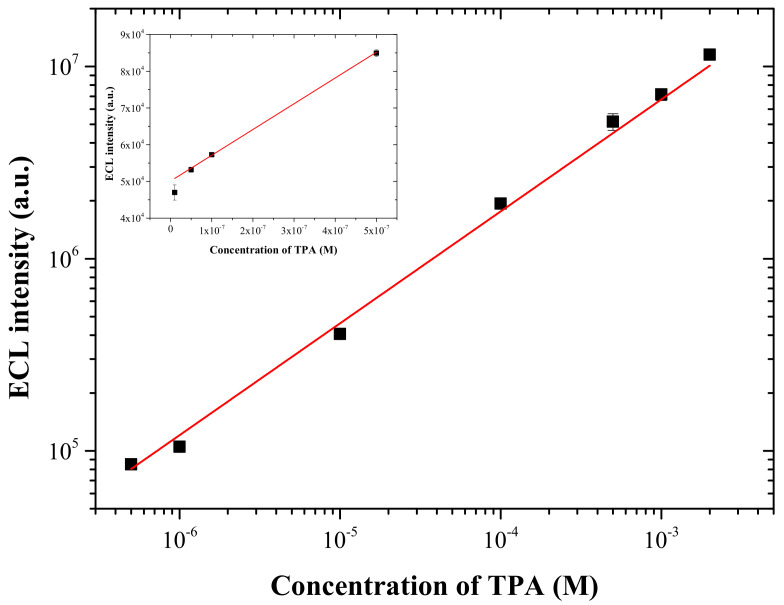
Calibration curve for TPA obtained at the Ru(bpy)_3_^2+^ ECL sensor based on the graphene-titania-Nafion composite in 0.05 M PBS at pH 7.0 with a scan rate of 100 mV/s. Inset: concentration range from 1.0 × 10*^−^*^8^ M to 5.0 × 10*^−^*^7^ M.

**Figure 8 sensors-22-03064-f008:**
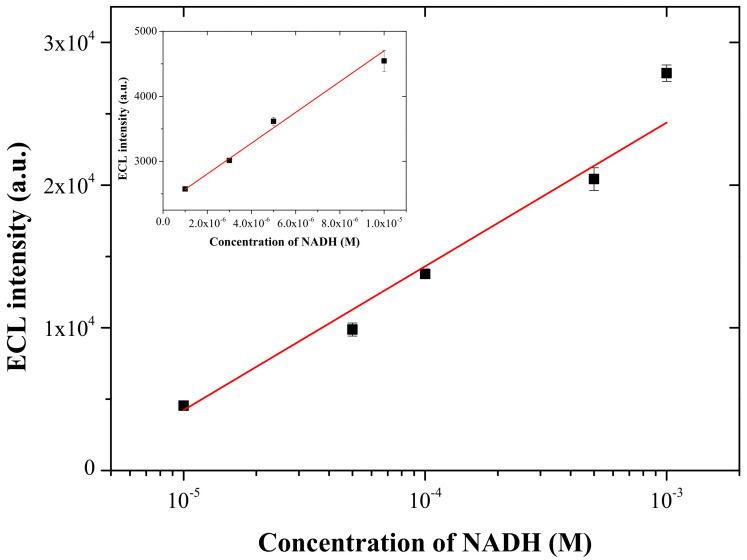
Calibration curve for NADH obtained at the present ECL sensor based on the graphene-titania-Nafion composite in 0.05 M PBS at pH 7.0 with a scan rate of 100 mV/s. Inset: concentration range from 1.0 × 10*^−^*^6^ M to 1.0 × 10*^−^*^5^ M.

**Table 1 sensors-22-03064-t001:** Comparison of the analytical performance of the Ru(bpy)_3_^2+^ ECL sensors based on different immobilization matrices.

Immobilization Matrix	Linear Range (M)	LOD (M)	Reference
Nafion	1.0 *×* 10^−6^–1.0 *×* 10^−3^	1.0 *×* 10^−6^	[5]
Titania-Nafion	1.0 *×* 10^−7^–1.0 *×* 10^−3^	1.0 *×* 10^−7^	[8]
CNT-Nafion	1.0 *×* 10^−7^–1.0 *×* 10^−3^	5.0 *×* 10^−8^	[16]
CNT-titania-Nafion	5.0 *×* 10^−8^–1.0 *×* 10^−3^	1.0 *×* 10^−8^	[17]
Graphene-Nafion	1.0 *×* 10^−7^–1.0 *×* 10^−4^	5.0 *×* 10^−8^	[19]
Graphene oxide	2.5 *×* 10^−7^–1.5 *×* 10^−4^	5.0 *×* 10^−9^	[20]
Graphene NS-PSS-C paste	5.0 *×* 10^−7^–1.0 *×* 10^−4^	4.0 *×* 10^−10^	[21,22]
Porous graphene-Nafion	5.0 *×* 10^−7^–1.0 *×* 10^−4^	4.0 *×* 10^−10^	[23]
Graphene-titania-Nafion	1.0 *×* 10^−8^–2.0 *×* 10^−3^	8.0 *×* 10^−10^	This study

Graphene NS: graphene nanosheets, PSS: poly(sodium4-styrenesulfonate).

**Table 2 sensors-22-03064-t002:** Selectivity study on ECL responses for various analytes.

Analytes (a)	Background-Corrected ECL Intensity (a.u.) (b)	Relative ECL Response (%)
Dopamine	6.80 *×* 10^2^	0.04
Tryptophan	1.71 *×* 10^3^	0.11
Ascorbic acid	1.79 *×* 10^3^	0.11
Sodium oxalate	3.82 *×* 10^3^	0.24
Ethanol	3.84 × 10^3^	0.24
Histidine	6.17 *×* 10^3^	0.39
Proline	8.24 *×* 10^4^	5.2
NADH	1.38 *×* 10^5^	8.7
Promazine	1.65 *×* 10^5^	10.4
Erythromycin	2.82 *×* 10^5^	17.8
TPA	1.58 *×* 10^6^	100

(a) The concentrations of all analytes were 0.5 mM in 0.05 M PBS. (b) Peak ECL intensities were measured on the present ECL sensor.

**Table 3 sensors-22-03064-t003:** Recovery test of the present ECL sensor for NADH in human serum samples.

Sample (a)	Added (µM)	Found (µM)	Recovery (%) (b)
1	0	2.4 ± 1.7	-
2	100	104.2 ± 6.3	104.2 ± 6.3
3	400	409.7 ± 5.2	102.4 ± 1.3

(a) Human serum sample was diluted 100 times. (b) Average of at least three determinations.

## Data Availability

Not applicable.

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
