# Peer review of "One-Step Fabrication of Highly Sensitive Tris(2,2′-bipyridyl)ruthenium(II) Electrogenerated Chemiluminescence Sensor Based on Graphene-Titania-Nafion Composite Film"

_sensors, 2022, doi:10.3390/s22083064_

Round 1
Reviewer 1 Report
In this manuscript, a highly sensitive tris(2,2’-bipyridyl)ruthenium(II) [Ru(bpy)32+] electrogenerated chemiluminescence (ECL) sensor based on graphene-titania-Nafion composite film was successfully prepared in a simple one-step manner. This one-step process for the construction of ECL sensor shortens the fabrication time and leads to reproducible ECL signals. Owing to the synergistic effect of conductive graphene and mesoporous titania-Nafion composite, the present ECL sensor leads to a highly sensitive detection of tripropylamine from 1.0*10-8 M to 2.0*10-3 M with a detection limit of 0.8 nM, which is one order of magnitude lower in comparison to previously reported ECL sensors based on CNT. Besides, the present ECL sensor has been also applied to the determination of NADH in human serum samples with good recovery. Although the novelty of this work in material design (TiO2, Nafion, Graphene) is not high, the good performance including selectivity and stability has been achieved based on the designed composite film. Overall, this work can be considered for publication in Sensors after the following revisions:
- Please use the full name of NADH in Abstract.
- The Title can be further improved as “One-Step synthesis of highly sensitive Tris(2,2’-bipyridyl)ruthenium(II) Electro-generated Chemiluminescence Sensor Based on Mesoporous Titania-Nafion-Graphene Composite Film”.
- The Introduction section can be further improved to clearly highlight the significance and novelty of this work.
- As for the structural and morphological features of the film, the material characterizations should be added, including XRD, SEM, TEM, Raman, and so on.
- To determine the stability of the sensor electrode, its microstructure and morphology should be characterized and compared before and after the long-term test.
- In Fig 5, please add the data of graphene-free titania-Nafion composite film.
- As for the CV plots, please give the area of sensor electrode and use the current density instead of current.
- As for EIS measurements, the information on the interfacial property at the graphene-titania-Nafion composite-modified GC electrode surface may be not related to the usage of [Fe(CN)6]3-/4- redox. Thus, please add more other EIS measurements in 0.5 mM TPA/0.05 M PBS and 0.05 M PBS.
Reviewer 2 Report
The manuscript “One-Step Prepared Tris(2,2’-bipyridyl)ruthenium(II) Electrogenerated Chemiluminescence Sensor Based on Mesoporous 3 Titania-Nafion Composite Film Containing Graphene” by Lee et al. reports a highly concentrated 0.1 M Ru(bpy)32+ solution was mixed with as-prepared graphene-titania-Nafion composite solution (1:20, v/v), and then a small aliquot (2 μL) of the resulting mixture solution was casted on a glassy carbon electrode surface. Although authors reported characterization of this sensor, some important results are still lack from this manuscript. Therefore, I would suggest authors may take at least a major revision before publication. Here are the comments and suggestions:
- In Fig. 5, why the ECL intensity is lower at 1.5 mg/mL of graphene than that at 1.25 or 1.75 mg/mL?
- Why the ECL intensity in Fig. 5 is lower than that in Figs. 4 or 6?
- What would be the shelf-life of this sensor?
- In Table 2, what would be the interfering of ethanol on this sensor?
- 6, 7 and their insets should be fitting with proper curves.
- In Table 3, the raw ECL intensities and the results without adding NADH in serum should be also reported.
Round 2
Reviewer 1 Report
The revised manuscript can be accepted for publication.
Reviewer 2 Report
It seems more acceptable now.